# Stereochemical Determination of Fistularins Isolated from the Marine Sponge *Ecionemia acervus* and Their Regulatory Effect on Intestinal Inflammation

**DOI:** 10.3390/md19030170

**Published:** 2021-03-22

**Authors:** Yeong Kwang Ji, Seon Min Lee, Na-Hyun Kim, Nguyen Van Tu, Yun Na Kim, Jeong Doo Heo, Eun Ju Jeong, Jung-Rae Rho

**Affiliations:** 1Department of Oceanography, Kunsan National University, Gunsan 54150, Korea; kwang7089@kunsan.ac.kr; 2Gyeongnam Department of Environment & Toxicology, Korea Institute of Toxicology, 17 Jegok-gil, Munsan-eup 52834, Korea; smlee84@kitox.re.kr (S.M.L.); nhkim@kitox.re.kr (N.-H.K.); jdher@kitox.re.kr (J.D.H.); 3Institute of Tropical Biology, 85 Tran Quoc Toan Street District 3, Ho Chi Minh 700000, Vietnam; nguyen.tu@itb.ac.vn; 4Department of Plant & Biomaterials Science, Gyeonsang National University, Jinju 52725, Korea; yunna@gntech.ac.kr

**Keywords:** fistularin, bromotyrosine alkaloid, *Ecionmeia acervus*, inflammatory bowel disease, co-culture, Mosher’s ester

## Abstract

By activity-guided fractionation based on inhibition of nitric oxide (NO) and prostaglandin E2 (PGE_2_), six fistularin compounds (**1**–**6**) were isolated from the marine sponge *Ecionemia acervus* (order Astrophorida). Based on stereochemical structure determination using Mosher’s method, fistularin-3 was assigned as a new stereoisomer. On the basis of the stereochemistry of fistularin-3, the stereochemical homogeneity of all six compounds was established by comparing carbon and proton chemical shifts. For fistularin-1 (**1**) and -2 (**2**), quantum calculations were performed to confirm their stereochemistry. In a co-culture system of human epithelial Caco-2 cells and THP-1 macrophages, all six isolated compounds showed potent anti-inflammatory activities. These bioactive fistularins inhibited the production of NO, PGE_2_, TNF-α, IL-1β, and IL-6 induced by lipopolysaccharide and interferon gamma. Inducible NO synthase and cyclooxygenase-2 expression and MAPK phosphorylation were downregulated in response to the inhibition of NF-κB nuclear translocation. Among the compounds tested, fistularin-1 (**1**) and 19-deoxyfistularin-3 (**4**) showed the highest activity. These findings suggest the potential use of the marine sponge *E. acervus* and its metabolites as pharmaceuticals for the treatment of inflammation-related diseases including inflammatory bowel disease.

## 1. Introduction

Bromotyrosine alkaloids are secondary metabolites derived from marine organisms, including marine sponges [1,2,3]. Bromotyrosine derivatives are divided into six categories: Simple bromotyrosine derivatives, spirocyclohexadienylisoxazolines, spirooxepinisoxazolines, oximes, bastadins, and other classes [4]. Fistularins are a family of brominated tyrosines found in various genera of marine sponges. Marine species that contain bromotyrosine alkaloids, including fistularins, are of particular interest because of their potent cytotoxicity. Fistularin-3 was first isolated from a specimen of *Aplysina fistularis* forma *fulva* in 1979 by Gopichand and Schmitz [5]. Since then, several fistularin family compounds have been isolated from the marine sponges, typically the order Verongida. Interestingly, fistularin-3 has been reported to have stereoisomers with different configurations at the two secondary carbinols C-11 and C-17, but initially not all stereoisomers were determined due to similar NMR spectra and crystal formation issues. Thereafter, although the configuration for C-17 was unassigned, Rogers et al. classified fistularin-3 into (+)-11(*S*) fistularin-3 and (+)-11-*epi*-fistularin-3 according to the absolute stereochemistry of C-11 by microscale LCMS-Marfey’s analysis [6]. In a recent study, 11-*epi*-Fistularin-3, isolated from the sponge *Suberea calvata*, was completely determined as (+)-1(*R*), 6(*S*), 1′(*R*), 6′(*S*), 11(*R*), 17(*S*) by the Mosher’s ester reaction and synergized with Bcl-2 inhibitor to kill acute myeloid leukemia cells [7]. Ciminiello et al. showed the different configuration for C-17 in 11-deoxyfistularin-3 by using the Mosher’s ester analysis, assigning as 17(*R*) [8]. In this study, a new stereoisomer of fistularin-3 was defined, which was identified as (+)-1(*R*), 6(*S*), 1′(*R*), 6′(*S*), 11(*S*), 17(*R*) by the modified Mosher’s ester reaction. Fistularin-3 and its derivatives showed cytotoxic activity [9]. 11-*epi*-Fistularin-3, a C-11 stereoisomer of fistularin-3 isolated from the tropical sponge *Agelas oroides*, was not cytotoxic to KB-cells (IC_50_ > 20 μg/mL), but 11-*epi*-Fistularin-3 has showed cytotoxicity towards cultured breast cancer cells (BC1 cells, IC_50_: 5.9 μg/mL; ZR-75-1 cells, IC_50_ = 4.5 μg/mL) [10]. Another derivatives of fistularin-3, 11-deoxyfistularin-3, isolated from the Caribbean sponge *Aplysina fistularis insularis,* showed cytotoxicity against cancer cells and most cytotoxic to MCF-7 human breast carcinoma cells, with an LD_50_ of 17 μg/mL [11]. Fitularin-3 was recently identified in a culture of *Pseudovibrio denitrificans* Ab134, a marine bacterium isolated from the sponge *Arenosclera brasiliensis*, suggesting the potential use of microorganisms to produce sustainable supplies of fistularin-3 and its derivatives for drug development [12].

We previously reported bioactive compounds from marine sponges that exerted anti-inflammatory activities useful to regulate intestinal inflammation in vitro [13,14] and in vivo [15]. During our ongoing screen for new metabolites with potential anti-inflammatory activity from marine sponges, we found that a methanolic extract of *Ecionemia acervus* significantly inhibited nitric oxide (NO) production in THP-1 macrophages. Besides stellatolides, a cyclodepsipiptide with cytotoxic activity against cancer cells, A549, HT-29, and MDA-MB-231 cells [16], few secondary metabolites from *E. acervus* have been reported. In our in vitro screening system using THP-1 macrophages activated with lipopolysaccharide (LPS) and interferon-gamma (IFNγ), the *E. acervus* extract showed no cytotoxicity at concentrations below 50 μg/mL. In addition, the ^1^H nuclear magnetic resonance (NMR) spectrum of the methanolic *E. acervus* extract and its organic fractions showed characteristic signals corresponding to fistularins. In this study, we attempted to identify the stereochemical fistularin derivatives from *E. acervus* and evaluated them using an in vitro co-culture system consisting of epithelial Caco-2 cells and differentiated THP-1 macrophages.

## 2. Results and Discussion

Using bioactivity-guided fractionation, six fistularins (**1**–**6**) were isolated from *E. acervus* (Figure 1). The structures of the six compounds were identified using NMR and high-resolution spectrometry (HR-MS): Fistularin-1 (FS-1, **1**), fistularin-2 (FS-2, **2**), fistularin-3 (FS-3, **3**), 17-deoxyfistularin-3 (17-deoxyFS-3, **4**) [9], 11-deoxyfistularin-3 (11-deoxyFS-3, **5**) [17], and 11,19-dideoxyfistularin-3 (11,19-dideoxyFS-3, **6**) [18].

### 2.1. Structures of Compounds ***1***–***6*** Isolated from Ecionemia acervus

Fistularin compounds were serially isolated from *E. acervus*. The structures of all compounds were established using a combination of HR-electron spray ionization-MS and NMR spectra. The circular dichroism (CD) spectra of all compounds were similar (Appendix A). Additional ^1^H and ^13^C NMR experiments for FS-3 (**3**) were conducted in acetone-*d*_6_ to compare with the NMR data for 11-*epi*-fistularin-3 (*epi*-FS-3) reported by Florean et al. [7] (Table 1, Appendix A). Table 1 shows that the proton and carbon chemical shifts for the two compounds (FS-3 and *epi*-FS-3) were nearly consistent, but the values of the proton and carbon chemical shifts at C-10, C-11, and C-18 of FS-3 were slightly smaller. For the purpose of determining a definite stereochemistry of **3**, the modified Mosher’s ester analysis was used, and the difference in the proton chemical shifts of the two esters was shown in Figure 2. The proton signals for (*S*)/(*R*)-tetraMTPA esters of **3** were assigned by the COSY spectrum (Appendix A), which enabled us to configure the four secondary carbon centers as 1*R*, 11*S*, 17*R*, and 1′*R*. Therefore, compound **3** was assigned as *SR*-FS-3 compared to the *RS*-FS-3 reported by Florean et al. [7]. Based on the stereochemical determination of **3** and comparison of the ^1^H and ^13^C chemical shifts of **3** with those of 17- deoxyFS-3 (**4**) and 11-deoxyFS-3 (**5**), the configurations of C-11 in **4** and C-18 in **5** could be assigned consistent values, indicating stereochemical homogeneity arising from the biogenetic biosynthesis of the same specimen. Compound **6** was proposed to be a biochemical precursor to **3**, **4**, and **5**, which involves hydroxylation of C-11 or C-17. The stereochemistry of **5** was supported by the match of the carbon chemical shifts of (17*R*)-11-deoxyFS-3 (**5**) with those reported in the literature [8].

In contrast, FS-1 (**1**) and -2 (**2**) possessed a 2-oxazolidone moiety on the one terminal, which is different from the pseudodimeric pattern of FS-3. Both compounds were deduced to be formed by esterification at each end of the chain in FS-3. Accordingly, the stereochemistry of these two compounds could be related to the stereochemical structure of FS-3. The configurations of the 2-oxazolidone moiety in **1** and **2** were suggested as 17*R* and 11*S*, respectively. Although the DP4+ method for **2** was not successful, the stereochemistry of the 2-oxazolidone moiety in **1** was supported by the calculation of DP4+ probability from the calculated and the measured NMR chemical shifts (Appendix A). After generating the initial structures for the two stereoisomers, **1a** (17*R*) and **1b** (17*S*), predicted by the values of proton coupling constants, each conformational search was performed by using molecular mechanics with a Merck molecular force field (MMFF), and six conformers with low energies (within a 10 kJ/mol threshold) were determined for the two stereoisomers. Geometry optimization of each conformer was performed at the B3LYP/6-31G(d) level, and then their single-point energies were calculated at the B3LYP/6-311+G(2d,p) level to obtain their respective Boltzmann populations at 298 K. ^1^H and ^13^C NMR chemical shifts for all conformers for the two stereoisomers were calculated at the MPW1PW97/6-311+G(p,d) level with the GIAO(Gauge Including Atomic Orbitals). Using DP4+ probability method [19], **1a** (17*R*) stereoisomer showed 100% probability, supporting the stereochemistry of **1**.

### 2.2. Inhibitory Activities of Compounds ***1***–***6*** on the Productioin of NO and PGE_2_ in THP-1 Macrophages Activated with LPS and IFNγ

Inhibitory effects of the isolated compounds (**1**–**6**) on production of NO and prostaglandin E2 (PGE_2_) were measured using Griess assay and enzyme-linked immunosorbent assay (ELISA), respectively (Figure 3). LPS and IFNγ are generally used stimuli to activate macrophages. Muller et al. [20] reported that the treatment of IFNγ was shown to synergize with TLR agonist, LPS, for induction of macrophage to produce NO and pro-inflammatory cytokines, TNF-α and IL-12. In our co-culture system, the concentration of LPS and IFNγ was optimized to activate THP-1 macrophages based on the increased production of NO and cytokines. The production of NO and PGE_2_ in THP-1 macrophages were significantly induced by co-treatment with LPS (10 μg/mL) and IFNγ (10 ng/mL). In CCK-8 cell viability assay, compound **2** showed the moderate cytotoxicity at concentrations greater than 5 μM, whereas the other compounds showed no cytotoxicity at concentrations below 20 μg/mL (cell viability > 98% of control cells). Based on cytotoxicity test, the concentrations of compounds were selected as 1.25, 2.5, and 5 μM for compound **2**; and 5, 10, and 20 μM for the other compounds. For NO and PGE2 assays, THP-1 cells were treated with each compound, followed by the addition of LPS + IFNγ. After 24 h of incubation, the contents of NO and PGE_2_ within the culture medium induced by LPS + IFNγ treatment were measured. The amounts of NO and PGE_2_ released in the culture medium decreased with increasing compound concentration. The levels of NO and PGE_2_ in culture media of cells pretreated with compounds **1**–**6** at the highest concentrations are summarized in Table 2. Compounds **1** and **2** were most potent on PGE_2_ inhibition, in which the contents of PGE_2_ were decreased to the level of non-treated control cells.

### 2.3. Compounds ***1***–***6*** Inhibit the Expression of Pro-Inflammatory Proteins iNOS and COX-2 in THP-1 Macrophages Co-Cultured with Caco-2 Cells

Prostaglandin (PG) levels are markedly elevated in the mucosa and rectal dialysate of patients with inflammatory bowel disease (IBD) [21,22]. Cyclooxygenase (COX) is the main enzyme that synthesizes PGs. In Crohn’s colitis and ulcerative colitis, but not in normal colon epithelium, COX-2 is expressed not only in epithelial cells, but also in mononuclear inflammatory cells. In addition to COX-2, iNOS reportedly is increased in epithelial cells in ulcerative colitis, but not in the normal colon [23]. Based on the inhibitory activities of compounds **1**–**6** on PGE_2_ in activated THP-1 macrophages, we investigated their regulatory effects on the expression of iNOS and COX-2 in an in vitro intestinal co-culture system. As shown in Figure 4, iNOS and COX-2 expression levels were significantly upregulated upon LPS + INFγ treatment. Compounds **1**–**6** significantly attenuated LPS + INFγ increases in iNOS and COX-2 expressed in a concentration-dependent manner. When comparing the inhibitory activities at the lowest concentration of 5 μM, both proteins were attenuated the most strongly in cells pretreated with compound **4**.

### 2.4. Compounds ***1***–***6*** Inhibit the Production of Pro-Inflammatory Cytokines in THP-1 Macrophages Co-Cultured with Caco-2 Cells

The expression of pro-inflammatory cytokines, including TNF-α, IL-1β, and IL-6, is increased in the intestinal lamina propria of patients with IBD [24]. In immune cells isolated from IBD patients, inflammatory mediators are present at markedly increased level when compared with those in normal tissues [25]. The IL-6 level is strongly increased in the colonic mucosa of patients with active IBD, and immune response products of anti-IL-6 antibody have been detected in infiltrating mononuclear cells in the lamina propria [26]. Increased levels of TNF-α have been reported in stool samples of children with active Crohn’s disease [27]. The effects of compounds **1**–**6** on the production of these inflammatory mediators were measured by ELISA. The increased production of IL-1β, IL-6, and TNF-α induced by LPS + IFNγ treatment was markedly suppressed in cells pretreated with compounds **1–6** (Figure 5). Cytokine levels cells treated with compounds **1–6** at the highest concentrations (5 μM for **2** and 20 μM for the others) are summarized in Table 3. The IL-1β level decreased the most strongly in cells treated with compound **4**.

### 2.5. Compounds ***1***–***6*** Inhibit MAPK Phosphorylation in THP-1 Macrophages Co-Cultured with Caco-2 Cells

Despite controversy regarding the therapeutic roles of mitogen-activated protein kinase (MAPK) inhibitors in IBD, several clinical trials have suggested a positive link between the inhibition of MAPKs and the treatment of IBD. In rodent models and human colonic biopsies, a specific inhibitor of JNK1/2 attenuated cytokine production and cell infiltration [28,29]. In IBD patients, sustained activation of ERK1/2 is observed. A recent preclinical study showed that a highly selective MEK inhibitor ameliorated murine colitis, demonstrating the potential beneficial effects of an inhibitor against ERK1/2 [30]. Glucocorticoids, a standard treatment for IBD, have been shown to inhibit ERK1/2 [31]. We evaluated the inhibitory effects of compounds **1**–**6** on the phosphorylation of MAPK family members, including p38, ERK1/2, and JNK (Figure 6). The increases in p38, ERK1/2, and JNK phosphorylation induced by LPS + IFNγ treatment were suppressed after pretreatment with compounds **1**–**6**. Compounds **3**–**5** showed the similar activities, with the inhibition of JNK phosphorylation being relatively weak when compared to the inhibition of ERK1/2 or p38 phosphorylation. Compounds **2** and **6** strongly inhibited ERK1/2 and JNK phosphorylation, whereas the inhibition of p38 phosphorylation was not significant. Compound **1** exhibited the most potent activity; p38, ERK1/2, and JNK phosphorylation was significant reduced in cells treated with compound **1** at concentrations higher than 10 μM.

### 2.6. Compounds ***1***–***6*** Inhibit the Nuclear Translocation of Nuclear Factor Kappa B (NF-κB) in THP-1 Macrophages Co-Cultured with Caco-2 Cells

NF-κB controls the transcription of inflammatory genes [32,33]. The primary mechanism of NF-κB is through inhibitory IκB proteins. Upon activation, NF-κB is rapidly released from its cytoplasmic inhibitor, IκB, and transmigrates into the nucleus. NF-κB binds to specific DNA elements, known as κB site, in gene promoter of target genes. Because the binding sites for NF-κB tare located proximally to the COX-2 and TNF-α promotors, inhibition of the DNA-binding activity of NF-κB resulted in the suppression of inflammatory mediators, including iNOS, COX-2, and TNF-α, in macrophages [24,25,34]. Considering the inhibitory effects of compounds **1**–**6** on the expression of pro-inflammatory cytokines and proteins, we investigated the effects of these compounds on localization of p65, which makes up the gene set of NF-κB, and on the phosphorylation of p65 in the cell nucleus and cytoplasm. In Western blot analysis, it was observed that compounds **1**–**6** down regulated the phosphorylation of p65 in the cytoplasm and inhibited the translocation to the nucleus (Figure 7). In addition, in cells pretreated with compounds, the total level of cytoplasmic IκB-α protein was significantly increased. Taken together, these results support that compounds **1**–**6** enhanced the binding of IκB-α to p65 resulting in the prevention of nucleus transloaction of NF-κB and consequent binding to the gene promoter. With respect to the translocation of p65 and cytoplasmic expression of IκB-α, compound **1** was found to be the most effective. Our results provide evidence that fistularins from *E. acervus* interfere with the interactions of the p65 subunit of NF-κB with specific sets of target genes, consequently inhibiting the transcription and expression of pro-inflammatory mediators [35,36].

## 3. Materials and Methods

### 3.1. General Experimental Procedures

Circular dichroism and optical rotation were measured using a Jasco J-715 spectropolarimeter and a JASCO P-1010 polarimeter (JASCO Corporation, Tokyo, Japan), respectively. SCIEX X500R (Sciex Co., Framingham, MA, USA) was used for HR-ESIMS experiment. All NMR spectra were recorded on Varian VNMRS 500 spectrometer (500 MHz for ^1^H, 125 MHz for ^13^C). Isolation and purification of compounds were performed on high-performance liquid chromatography (HPLC) system equipped with an YMC ODS-A column (250 × 10 mm, 5 μm) or a Phenomenex C8 column (250 × 10 mm, 5 μm). Dell PowerEdge R740 Server (Dell, Rounf Rocks, TX, USA) was installed for DFT NMR calculations. Gaussian 16 (Gaussian. Inc., Wallingford, CT, USA) and Spartan’18 software (Wavefunction Inc, Irvine, CA, USA) was used for.

### 3.2. Chemicals and Antibodies

All solvents for isolation were used after distillation. NMR solvents (Acetone-*d6*, D_2_O, CD_3_Cl_3_, CD_3_OD) and chemicals [dimethylaminopyridine, *R*(–)-MTPA-Cl, *S*(+)-MTPA-Cl] were purchased from Sigma-Aldrich (St. Louis, MO, USA).

MEM, RPMI 1640, LPS (L4516, *Escherichia coli* 0127: B8), and PMA (2-mercaptoethanol, phorbol 12-myristate 12-acetate) were purchased from Sigma-Aldrich (St. Louis, MO, USA). Fetal bovine serum (FBS) and penicillin/streptomycin were purchased from Gibco Life Technologies (Grand Island, NY, USA). The Griess Reagent System was from Promega (Promega, Madison, WI, USA). ELISA kits for TNF-α, IL-6, and PGE_2_ were from R&D Systems (Minneapolis, MN, USA). Anti-iNOS, anti-COX-2, anti-p65, anti-lamin B, anti-phospho-p65, anti-IκBα, anti-phospho-ERK, anti-ERK, anti-phospho-p38, anti-p38, anti-phospho-JNK, anti-JNK, anti-HO-1 and anti-Nrf2 were purchased from Cell Signaling Technology, Inc. (Danvers, MA, USA). Anti-β-actin and secondary antibodies were purchased from Sigma-Aldrich (St. Louis, MO, USA). 

### 3.3. Collection of Marine Sponge, E. acervus

Marine sponge, *E. acervus* (phylum Porifera, class Demospongiae, order Astrophorida, family Ancorinidae) was collected in Vietnam (09°54′40.9″ N, 104°01′44.2″ E) in April 2018 and immediately frozen. A voucher specimen (MABIK Lot No. 0014011) was deposited at the Marine Biodiversity Institute of Korea (MABIK) and was identified by Dr. Young-A Kim, a professor of Hannam University. Frozen samples of *E. acervus* were stored at −25 °C until use.

### 3.4. Isolation of Fistularins from E. acervus

The specimen (4.5 kg) was extracted with MeOH (2 L × 2) at 25 °C for 24 h and partitioned into H_2_O and CH_2_Cl_2_ fractions for desalting. CH_2_Cl_2_ soluble fraction (3.8 g) was subjected to ODS silica gel open-column chromatography with a stepwise elution to yield six fractions (50% H_2_O (I)→100% MeOH (VI) with gradual increments of 10% MeOH). Fraction III (190 mg) and fraction V (230 mg) that showed significant NO inhibitory activity were further subjected to size exclusive chromatography to yield five subfractions (M1 to M5). Subfraction M4 (29 mg) from fraction III was isolated using HPLC equipped with UV detector (Phenomenex polar, 250 × 10 mm, 2 mL/min) with a gradient elution of H_2_O and MeOH to give compounds **1** (1.3 mg) and **2** (2.3 mg). Subfraction M5 of fraction V was isolated using HPLC equipped with RI detector (Phenomenex C8, 250 × 10 mm, 2 mL/min) with a gradient elution of 30% H_2_O + 70% MeOH to give compounds **3** (25 mg), **4** (2.3 mg), **5** (3.2 mg), and **6** (4.4 mg).

Fistularin-1 (**1**): an amorphous white solid; ${\left[\alpha \right]}_{D}^{25}$ = + 65.2 (*c* 0.1, MeOH); ^1^H and ^13^C NMR see Appendix A.

Fistularin-2 (**2**): an amorphous white solid; ${\left[\alpha \right]}_{D}^{25}$ = + 41.6 (*c* 0.1, MeOH); ^1^H and ^13^C NMR see Appendix A.

Fistularin-3 (**3**): an amorphous white solid; ${\left[\alpha \right]}_{D}^{25}$ = + 151.6 (*c* 0.1, MeOH); ^1^H and ^13^C NMR see Appendix A.

19-deoxyfistularin-3 (**4**): an amorphous white solid; ${\left[\alpha \right]}_{D}^{25}$ = + 62.0 (*c* 0.1, MeOH); ^1^H and ^13^C NMR see Appendix A.

11-deoxyfistularin-3 (**5**): an amorphous white solid; ${\left[\alpha \right]}_{D}^{25}$ = + 55.6 (*c* 0.1, MeOH); ^1^H and ^13^C NMR see Appendix A.

11,19-dideoxyfistularin-3 (**6**): an amorphous white solid; ${\left[\alpha \right]}_{D}^{25}$ = + 51.6 (*c* 0.1, MeOH); ^1^H and ^13^C NMR see Appendix A.

### 3.5. Mosher’s Reaction of FS-3

Dried FS-3 (2.0 mg) was dissolved in acetone-*d*_6_ (500 μL), and the ^1^H NMR spectrum was measured. After adding dimethylaminopyridine (DMAP, 3.5 mg) into the NMR cell and vortexing, the ^1^H NMR spectrum of the mixture was measured. Then, the mixture was divided into two aliquots. One fraction was mixed with *R*(–)-MTPA-Cl (4 μL) in an NMR cell, and the other was mixed with *S*(+)-MTPA-Cl (7 μL) in another NMR cell to monitor the reaction. After the NMR signals were stabilized, each solution was dried and partitioned into D_2_O and CDCl_3_ solvents. The chloroform fraction was used for the NMR experiments.

(*S*)-MTPA ester of **3**: ^1^H NMR (500 MHz, CDCl_3_) δ 7.47 (2H, s, H-15, 15′), 6.71 (1H, m, NH), 6.59 (1H, m, NH’), 6.42/6.40 (2H, s, H-1, 1′), 6.12/6.11 (2H, s, H-5, 5′), 5.95 (1H, dd, *J* = 9.3, 3.2 Hz, H-17), 5.60 (1H, m, H-11), 4.25 (1H, dd, *J* = 10.3, 3.7 Hz, H-12), 4.21 (1H, dd, *J* = 10.3, 5.1 Hz, H-12), 3.98 (1H, dd, *J* = 14.7, 4.7 Hz, H-10), 3.87 (1H, dd, *J* = 14.7, 3.4 Hz, H-18), 3.77/3.76 (6H, s, OCH_3_), 3.66 (1H, dd, *J* = 14.4, 7.1 Hz, H-10), 3.51 (2H, dd, *J* = 18.3, 9.1 Hz, H-7, 7′), 3.26 (1H, dd, *J* = 14.4, 4.9 Hz, H-18), 3.06 (2H, dd, *J* = 18.3, 7.3 Hz, H-7, 7′).

(*R*)-MTPA ester of **3**: ^1^H NMR (500 MHz, CDCl_3_) δ 7.32 (2H, s, H-15, 15′), 6.88 (1H, m, NH), 6.67 (1H, m, NH’), 6.28 (2H, s, H-1, 1′), 6.03/6.01 (2H, s, H-5, 5′), 5.83 (1H, dd, *J* = 9.5, 3.4 Hz, H-17), 5.57 (1H, m, H-11), 4.20 (2H, d, *J* = 4.2 Hz, H-12), 4.13 (1H, m, H-10), 3.86 (1H, m, H-18), 3.79 (1H, m, H-10), 3.79/3.78 (6H, s, OCH_3_), 3.33 (1H, dd, *J* = 15.9, 9.8 Hz, H-18), 3.27 (2H, dd, *J* = 18.6, 12.0 Hz, H-7, 7′), 2.89 (2H, dd, *J* = 18.6, 5.1 Hz, H-7, 7′).

### 3.6. DP4+ Probability of Compound ***1***

Two plausible isomers for compound **1** were assumed to be **1a** (17*R*) and **1b** (17*S*). By using the MMFF force field and energy minimization, six conformers for each isomer were explored within relative energies below 10 KJ/mol. All conformers were optimized at the B3LYP level with 6-31G(d) basis set, and further calculations of the ^1^H and ^13^C NMR chemical shifts for each conformer were performed at the mPW1PW91/6-311+G(d,p) level with the polarizable continuum model in MeOH. The ^1^H and ^13^C NMR chemical shifts for **1a** and **1b** were calculated with a Boltzmann distribution for the DP4+ probability. DP4+ calculation was conducted with excel file given by Ref. [19].

### 3.7. Cell Cultures

Caco-2 cells (human epithelial) and THP-1 cells (human monocytic) were purchased from the American Type Culture Collection (ATCC, Manassas, VA, USA). Caco-2 cells were cultured in MEM with 10% FBS and 1% penicillin- streptomycin. THP-1 cells were cultured in RPMI1640 with 10% FBS and 1% penicillin- streptomycin and 0.05 mM 2-mercaptoethanol. The cells were incubated in a humidified incubator at 37 °C supplied with atmosphere of 95% and-5% CO_2_.

### 3.8. Differentiation of THP-1 to Macrophages

Monocytic THP-1 cells were differentiated with PMA. After incubation of cells in culture medium containing PMA (50 ng/mL) for 3 days, the medium was replaced with fresh one without PMA. Cells were further incubated for 2 days.

### 3.9. In Vitro Co-Culture Model of Intestine

To establish the human intestinal co-culture system, Caco-2 cells were seeded on transwell inserts (pore size: 0.4 μm; Corning CoStar Corp., Cambridge, MA, USA) at the density of 3.75 × 10^5^ cells/well and maintained in an incubator for 14–20 days. The culture medium was changed every three days until the cells were fully differentiated (checked by transepithelial electrical resistance TEER value > 1200 Ω cm^2^). THP-1 cells were independently seeded onto the plate bottom of a 6-well transwell plate at 8.5 × 10^6^ cells/well. When Caco-2 cells were fully differentiated, transwell insert with Caco-2 monolayer was added. The upper and lower chambers would represent the apical and basolateral sides of the intestinal epithelium, respectively [37,38]. To evaluate anti-inflammatory activity, 10 µg/m of LPS and 10 ng/mL of IFNγ were treated to the apical compartment of the plate to stimulate THP-1 indirectly through Caco-2 monolayer, and the test compounds to be tested were added to the basolateral compartment to evaluate direct anti-inflammatory activities against THP-1. THP-1 cells and the culture supernatant from the basolateral side were collected for western blotting and measurement of inflammatory mediators.

### 3.10. Measurment of TEER Values

TEER value was measured using a Millicell-ERS instrument (Millipore, MA, USA) to assess integrity of Caco-2 monolayer. TEER values were calculated as follows: TEER(Ω∙cm^2^) = Resistance − blank resistance (Ω) × membrane surface area (cm^2^). TEER values are extensively used in co-culture model development to measure the resistance of the tight junctions of cell monolayers [38]. In our co-culture system, TEER value of the Caco-2 layer significantly increased for first four days and reached a maximum on day 21.

### 3.11. Cytotoxicity Assay

Cytotoxicity of each compound was measured using the Cell Counting Kit-8 (CCK-8; Donjinjo Molecular Technologies, Inc., Kumamoto, Japan), according to the manufacturer’s instructions. THP-1 cells were seeded at 1 × 10^4^ cells/well in a 96-well plate and incubated for 24 h. The cells were treated with each compound at various concentrations. After the incubation of 24 h, 10 μL of CCK-8 solution was added to each well and incubated for 3 h at 37 °C. The absorbance at 450 nm was measured using a microplate reader (Bio-Tek Company, Winooski, VT, USA).

### 3.12. Measurement of NO and Inflammatory Cytokines

THP-1 cells were seeded in 96-well plates at 1 × 10^4^ cells/well and incubated overnight. The cells were treated with samples to be tested, and then treated with LPS (10 µg/mL) and IFNγ, (10 ng/mL) for 24 h. The amount of NO produced within the culture medium was determined using Griess reagent assay. The amount of PGE_2_ in culture medium, and the amount of IL-1β, TNF-α, and IL-6 in the culture medium from the basolateral compartment of the Transwell plate were measured using ELISA kits (R&D Systems, Minneapolis, MN, USA), according to the manufacturer’s instructions. The Absorbance was measured using a microplate reader (Bio-Tek Instruments).

### 3.13. Western Blotting

The co-culture system was established as described in Section 3.9. After incubation with test compounds and/or LPS + IFNγ, THP-1 cells were collected and lysed with a RIPA buffer containing a protease inhibitor cocktail (Santa Cruz Biotechnology, Santa Cruz, CA, USA) on ice for 40 min. Protein lysates were centrifuged at 13,000× *g* at 4 °C for 30 min. Protein samples (30 μL) were separated by sodium dodecyl sulfate-polyacrylamide gel electrophoresis (8–12%) at 100 V and transferred to polyvinylidene difluoride membranes. The membrane was blocked with 5% non-fat milk in a PBST buffer and incubated with the specific primary antibodies at 4 °C overnight. Proteins of interest was detected with secondary antibodies at room temperature for 1 h. Bands were visualized using ECL solution (Thermo Fisher Scientific) and quantified using the Chemidoc Imaging System (Bio-Rad, Hercules, CA, USA).

### 3.14. Statistical Analysis

Data were analyzed using Prism version 5.00 (GraphPad Software, San Diego, CA, USA). Significant differences between the two groups were analyzed using Student’s *t*-test and one-way ANOVA. *p* < 0.05 was considered as statistically significant. Data from three independent experiments are expressed as the mean ± SD.

## 4. Conclusions

Most of bromotyroine secondary metabolites, including fistularins, have been isolated from the sponges of the order Verongida sufficient for use as a chemical marker [4,39]. In this study, six fistularin compounds (**1**–**6**) were isolated from *E. acervus* (order Astrophorida) collected in Vietnam and showed a deviating chemotaxonomy. Specifically, fistularin-3 (**3**), which possesses four secondary hydroxy groups, has been isolated as different stereoisomers depending on the sponge genus and the collection site [40,41]. The complete absolute stereochemistry of fistularin-3 (**3**) isolated in this study was determined as 1(*R*), 6(*S*), 1′(*R*), 6′(*S*), 11(*S*), 17(*R*), assigning a new stereoisomer, *SR*-FS-3. Comparison of NMR spectral data and an electronic CD spectrum did not provide clear evidence of the stereochemistry of **3**. Instead, the Mosher’s method allowed assigning the configuration of the four chiral carbons with a secondary hydroxy group. Based on a comparison of the NMR chemical shifts with the determined structure of **3** and hypothesis of biogenetic biosynthesis, the configurations of the other derivatives were established. The anti-inflammatory effects of compounds **1**–**6** were evaluated in a co-culture system established using epithelial Caco-2 cells and THP-1 macrophages to mimic the intestinal environment. Fistularin derivatives from *E. acervus* inhibited the production of NO and pro-inflammatory cytokines and regulated the expression of iNOS and COX-2, as well as the MAPK family (ERK1/2, p38, JNK). Nuclear translocation of NF-κB was also attenuated. Although various differences in activity were observed, all compounds showed the potent anti-inflammatory activities in the intestinal co-culture system. Our findings support the potential use of the marine sponge *E. acervus* and its bioactive secondary metabolites, fistularins, as therapeutic agents in the treatment of IBD.

## Figures and Tables

**Figure 1 marinedrugs-19-00170-f001:**
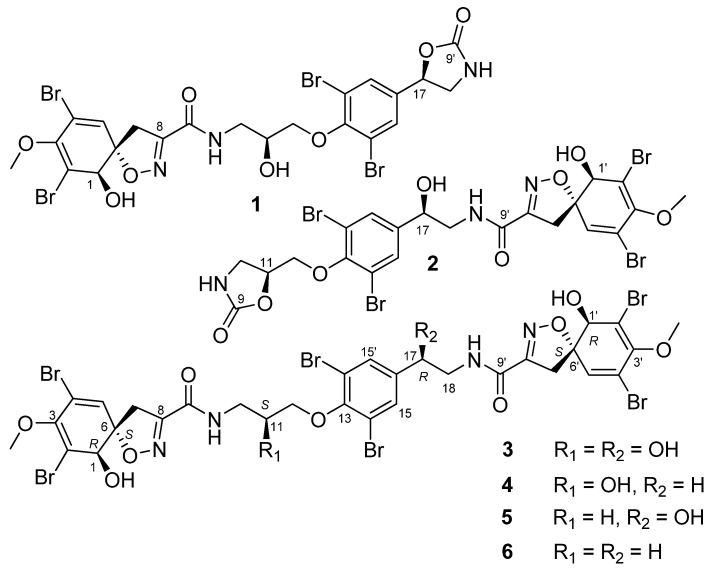
The structures of compounds **1**–**6** isolated from the marine sponge *Ecionemia acervus.*

**Figure 2 marinedrugs-19-00170-f002:**
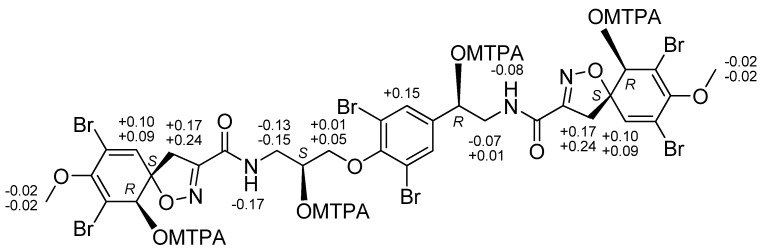
Difference (Δδ*^SR^*) in ^1^H NMR chemical shifts for *S*/*R*-MTPA esters of **3** in CDCl_3_ solvent.

**Figure 3 marinedrugs-19-00170-f003:**
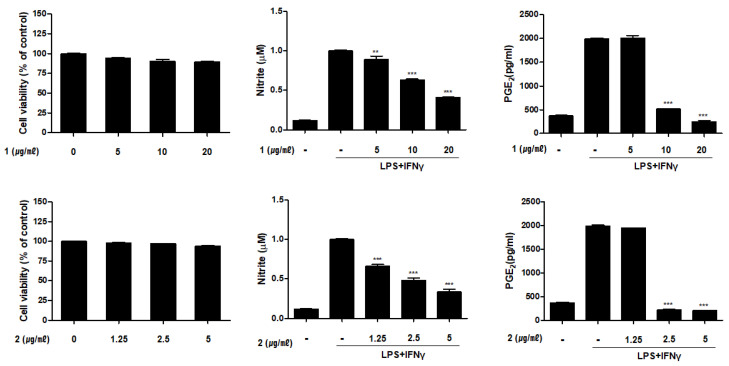
Compounds **1**–**6** from *E. acervus* inhibited the production of nitric oxide (NO) and prostaglandin E2 (PGE_2_) in THP-1 cells activated with lipopolysaccharide (LPS) + interferon-gamma (IFNγ). THP-1 cells were treated with various concentration of compound **1**–**6**, followed by treated with LPS (10 μg/mL) + IFNγ (10 ng/mL). The levels of NO and PGE_2_ in culture medium were analyzed using Griess assay and ELISA kit, respectively. Results are shown as the mean ± SD; ** *p* < 0.01 and *** *p* < 0.001 compared to LPS+IFNγ only treated cells (*n* = 3).

**Figure 4 marinedrugs-19-00170-f004:**
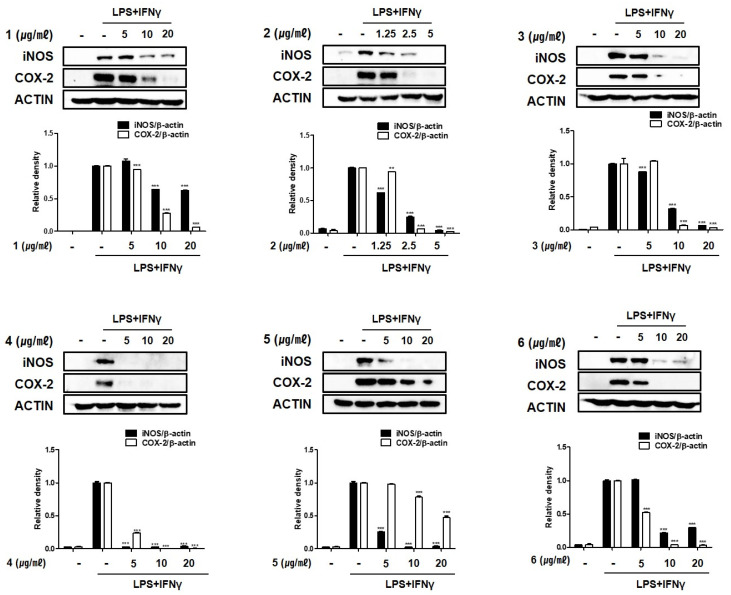
Compounds **1**–**6** from *E. acervus* inhibited the expression of iNOS and cyclooxygenase (COX-2) in THP-1 cells co-cultured with Caco-2 cells. LPS + IFNγ was added into the apical compartment, and various concentration of compounds **1**–**6** was added to the basolateral compartment of the Caco-2/THP-1 co-culture system. After 24 h of incubation, the levels of iNOS and COX-2 proteins were analyzed using Western blotting. Results are shown as the mean ± SD; ** *p* < 0.01 and *** *p* < 0.001 compared to LPS + IFNγ only treated cells (*n* = 3).

**Figure 5 marinedrugs-19-00170-f005:**
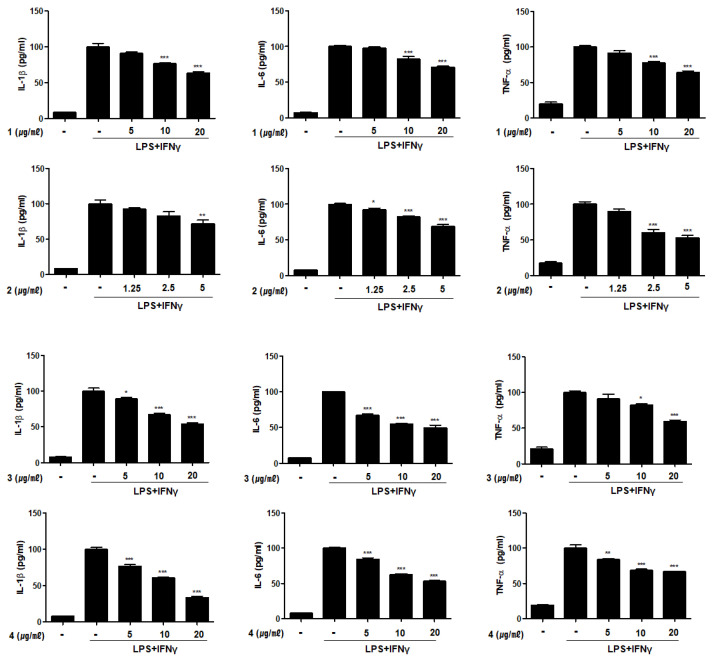
Compounds **1**–**6** from *E. acervus* inhibited the production of inflammatory cytokines in THP-1 cells co-cultured with Caco-2 cells. LPS + IFNγ was added into the apical compartment, and various concentration of compounds **1**–**6** was added to the basolateral compartment of the Caco-2/THP-1 co-culture system. After 24 h of incubation, the levels of IL-1β, IL-6, and TNF-α in the culture medium were analyzed using ELISA kit. Results are shown as the mean ± SD; * *p* < 0.05, ** *p* < 0.01 and *** *p* < 0.001 compared to LPS + IFNγ only treated cells (*n* = 3).

**Figure 6 marinedrugs-19-00170-f006:**
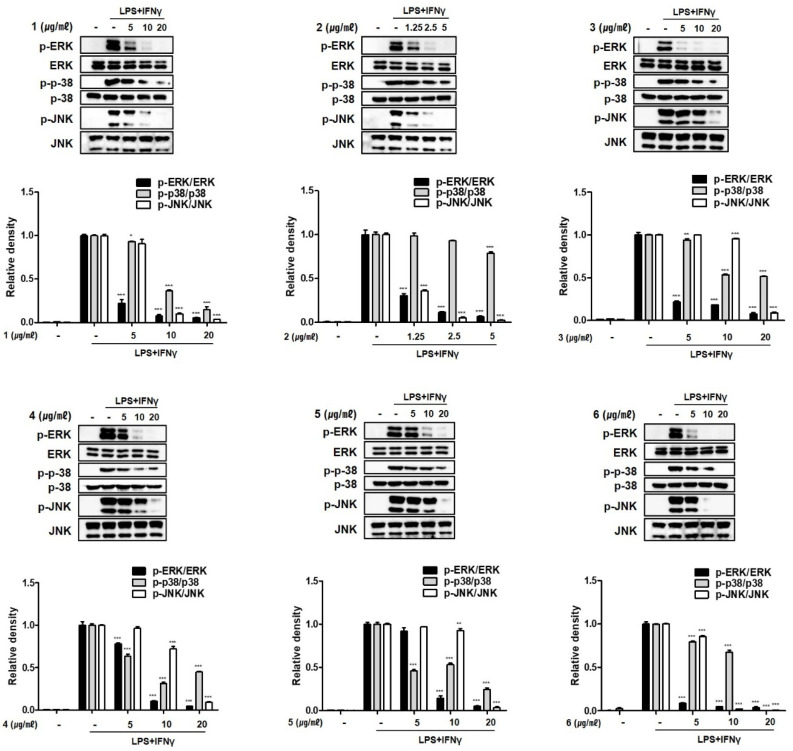
Compounds **1**–**6** from *E. acervus* inhibited the MAPKs phosphorylations in THP-1 cells co-cultured with Caco-2 cells. LPS + IFNγ was added into the apical compartment, and various concentration of compounds **1**–**6** was added to the basolateral compartment of the Caco-2/THP-1 co-culture system. After 24 h of incubation, the phosphorylation levels of ERK, p38 and JNK in cells were analyzed using Western blotting. Results are shown as the mean ± SD; * *p* < 0.05, ** *p* < 0.01 and *** *p* < 0.001 compared to LPS + IFNγ only treated cells (*n* = 3).

**Figure 7 marinedrugs-19-00170-f007:**
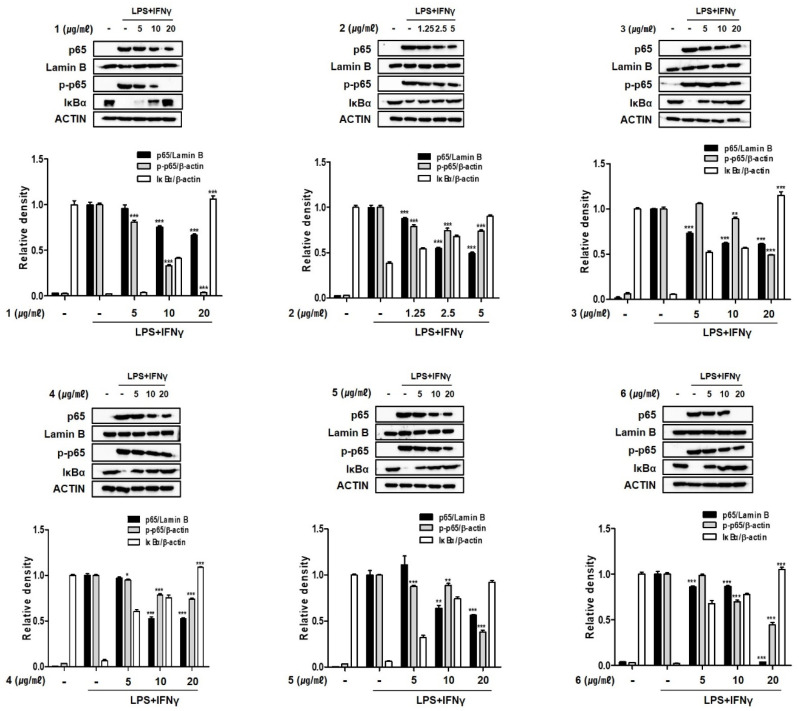
Compounds **1**–**6** from *E. acervus* inhibited nuclear translocation of NF-κB in THP-1 cells co-cultured with Caco-2 cells. LPS + IFNγ was added into the apical compartment, and various concentration of compounds **1**–**6** was added to the basolateral compartment of the Caco-2/THP-1 co-culture system. After 24 h of incubation, the expression of each protein was analyzed using Western blotting. Nuclear fraction was normalized to Lamin B, cytosol fraction and whole cell lysate were normalized to β-actin. Results are shown as the mean ± SD; * *p* < 0.05, ** *p* < 0.01 and *** *p* < 0.001 compared to LPS + IFNγ only treated cells (*n* = 3).

**Table 1 marinedrugs-19-00170-t001:** The nuclear magnetic resonance (NMR) Spectral data for FS-3 (**3**) and *epi*-FS-3 (acetone-*d*_6_).

No	FS-3 (11*S*, 17*R*)		*epi*-FS-3 (11*R*, 17*S*) *	
	^13^C	^1^H	^13^C	^1^H
1, 1′	75.1, 75.1	4.17, s/4.19, s	75.2, 75.3	4.18, s/4.19, s
2, 2′	122.1, 122.2		122.1, 122.1	
3, 3′	148.7, 148.7		148.8, 148.8	
4, 4′	113.8, 113.8		113.8, 113.9	
5, 5′	132.2, 132.3	6.53, s/6.55, s	132.3, 132.4	6.52, s/6.53, s
6, 6′	91.3, 91.3		91.8, 91.8	
7, 7′	39.5, 39.5	3.21/3.86, d (18.1)	40.0, 40.0	3.19 /3.85, d (18.0)
		3.18/3.83, d (18.1)		3.16 /3.82, d (18.0)
8, 8′	155.1, 155.1		155.1, 155.2	
9, 9′	160.3, 160.4		160.5, 160.5	
10	43.4	3.52, dd (13.9, 7.3)	43.6	3.54, m
		3.78, dd (13.7, 4.7)		3.80, m
11	69.7	4.25, m	69.9	4.25, m
12	75.8	4.02, dd (9.1, 5.9)	75.9	4.02, dd (9.1, 5.5)
		4.07, dd (9.1, 5.6)		4.06, dd (9.1, 5.5)
13	152.6		152.7	
14, 14′	118.3, 118.3		118.4, 118.4	
15, 15′	131.4, 131.4	7.67, s	131.5, 131.5	7.66, s
16	143.3		143.3	
17	71.3	4.90, dd (7.3, 4.7)	71.3	4.90, dd (7.7, 4.3)
18	47.4	3.48, dd (13.7, 7.3)	47.7	3.49, m
		3.60, dd (13.7, 4.7)		3.63, m
OCH_3_	60.2	3.73, s	60.2	3.73, s
NH		7.69, br t (5.4)		7.62, br t (6.0)
NH’		7.75, br t (6.1)		7.66, br t (6.0)

* From Ref. [7].

**Table 2 marinedrugs-19-00170-t002:** Inhibitory effects of compounds **1**–**6** on the production of NO and PGE_2_ in THP-1 cells activated with LPS + IFNγ. Values are presented as the means ± SD (*n* = 3).

Compounds (20 μM)	NO	PGE_2_
	Relative %	
Non-treated control	0.00	
LPS + IFNγ only	100.00	
1	32.97 ± 0.85	−7.50 ± 3.03
2 *	24.46 ± 7.42	−10.40 ± 0.39
3	19.39 ± 0.66	15.54 ± 1.95
4	39.82 ± 4.94	3.70 ± 1.61
5	33.24 ± 3.09	67.98 ± 1.60
6	33.22 ± 2.65	2.63 ± 0.51

* Compound **2** was used at 5 μM, the other compounds at 20 μM.

**Table 3 marinedrugs-19-00170-t003:** Inhibitory effects of compounds **1**–**6** on the production of anti-inflammatory cytokines in THP-1 macrophages co-cultured with Caco-2 cells. Values are presented as the mean ± SD (*n* = 3).

Compounds (20 μM)	IL-1β	IL-6	TNF-α
	Relative %		
Non-treated control	0.00		
LPS+ IFNγ only	100.00		
1	58.62 ± 3.00	67.32 ± 2.13	58.97 ± 4.75
2 *	67.66 ± 11.00	64.89 ± 5.23	46.84 ± 6.21
3	48.17 ± 3.00	42.94 ± 6.77	54.21 ± 3.46
4	25.05 ± 3.41	46.96 ± 1.14	62.03 ± 1.06
5	34.37 ± 2.96	48.25 ± 0.62	48.19 ± 3.47
6	60.07 ± 2.95	64.82 ± 1.60	50.83 ± 2.47

* Compound **2** was used at 5 μM, the other compounds at 20 μM.

## Data Availability

Not Applicable.

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
