# Peer review of "Stereochemical Determination of Fistularins Isolated from the Marine Sponge Ecionemia acervus and Their Regulatory Effect on Intestinal Inflammation"

_marinedrugs, 2021, doi:10.3390/md19030170_

Round 1
Reviewer 1 Report
The work is devoted to the establishment of the chemical structure and the study of biological activity of a number of bromotyrosine alkaloids, fistularins, isolated from the sea sponge Ecionemia acervus collected in Vietnam waters.
Using HPLC, 1H and 13C NMR methods and quantum calculations, the authors established the complete stereochemical structures of six new compounds.
Using an in vitro model of colitis based on a co-cultivation of human epithelial and macrophagal cells, inflammation in which was caused by standard inducers of inflammation LPS+IFNgamma, the authors demonstrated the effect of the obtained compounds. Fistularins have been identified that strongly inhibit the production of some inflammatory biomarkers, such as NO, PGE2, TNF-α, IL-1β, and IL-6, as well as blocking the expression of iNOs and COX-2, and inhibition of MAPK phosphorylation and NF-κB nuclear translocation involved in inflammation process.
The data obtained indicate the prospects for further study of the anti-inflammatory properties of the new series of bromotyrosine alkaloids as anti-inflammatory agents.
The work will be of interest to readers and may be published in Marine Drugs.
Notes:
1. The results of the cytotoxic activity of bromotyrosine alkaloids are presented only in relation to THP-1 macrophages. Since most of the work on the anti-inflammatory activity was performed on THP-1 macrophages co-cultured with Caco-2 cells, the range of cytotoxic concentrations of the studied fistularins against Caco-2 cells should also be indicated in the text.
2. Tables 2 and 3 duplicate Figures 3 and 5, respectively. It is better to remove the tables, and indicate the percentages of the inhibitory effect of the most active compounds in the manuscript text.
3. The order of numbering of figures and references to them in the text is violated:
- There is no reference to Figure 3 in the text;
- Line 175 - the reference to Figure 5 is incorrect. There should be Figure 4;
- Line 202 - change the reference from Figure 6 to Figure 5;
- Line 230 - change the reference from Figure 5 to Figure 6;
- Line 260 - reference to non-existent Figure 7;
4. Line 418 - give the city and country for R&D Systems manufacturer;
5. Line 423 - Biotek should be Bio-Tek;
6. Line 425 - The co-culture system is not described in section 3.6.
Author Response
Dear reviewer,
We thank you for providing the helpful and insightful comments to improve our manuscript. Our point-by-point response to your comments is provided below. The modified parts are marked in Red. Also, we notice that the manuscript has been carefully edited for spelling, grammar, and readability using Editage’s language editing service.
- The results of the cytotoxic activity of bromotyrosine alkaloids are presented only in relation to THP-1 macrophages. Since most of the work on the anti-inflammatory activity was performed on THP-1 macrophages co-cultured with Caco-2 cells, the range of cytotoxic concentrations of the studied fistularins against Caco-2 cells should also be indicated in the text.
→ Similarly to results in THP-1 cells, compounds 1~7 showed no cytotoxicity against Caco-2 cells in co-culture system at the concentration applied in all assays. This was additionally described in the text. Section 2.2 has been revised.
- Tables 2 and 3 duplicate Figures 3 and 5, respectively. It is better to remove the tables, and indicate the percentages of the inhibitory effect of the most active compounds in the manuscript text.
→ Thank you for your comments. We know that Tables 2 and 3 duplicate Figures 3 and 5. Nevertheless, we authors decided to leave Tables 2 and 3 in the text, because we believe that presenting the results in a table is helpful for readers to see the potency of each compound at the highest concentration, and also is better to improve readers' understanding and readability.
- The order of numbering of figures and references to them in the text is violated:
- There is no reference to Figure 3 in the text;
- Line 175 - the reference to Figure 5 is incorrect. There should be Figure 4;
- Line 202 - change the reference from Figure 6 to Figure 5;
- Line 230 - change the reference from Figure 5 to Figure 6;
- Line 260 - reference to non-existent Figure 7;
→ Thank you for your careful review for our manuscript. The numbering and citation of all figures are revised.
- Line 418 - give the city and country for R&D Systems manufacturer;
→ the city and country for R&D system is added
- Line 423 - Biotek should be Bio-Tek;
→ Bioteck is corrected to Bio-Tek
- Line 425 - The co-culture system is not described in section 3.6.
→ 3.6 was corrected to 3.8 in the text
Reviewer 2 Report
This manuscript describes the activity guided re-isolation of several fistularin-type compounds, and the full assignment of their stereochemistry using Mosher ester analysis, as well as several subsequent biological studies. It would appear that this is the first time that the C-17 stereochemistry of fistularin 3 has been conclusively defined, which represents novelty and a significant advancement. In most places, the paper is well-written, logically presented and the conclusions are well-supported. However, prior to publication, the manuscript could definitely benefit from clarifications in a few places. Specifically, compared to the rest of the manuscript which is quite strong, the authors do a very poor job of bringing the reader up-to-speed in terms of what is known and unknown about the C-11/C-17 fistularin isomers which have been isolated and reported in the literature. Some point-by-point examples of this issue are provided below:
Lines 46-49 - The statements here are self-contradictory. Regarding fistularin-3, is first stated that "its absolute stereochemistry has not been determined because the absolute configurations of the secondary carbinol groups C-11 and C-17 have not been defined," and in the next sentence it stated that Rogers/Molinski determined the absolute stereochemistry of the compound and its C-11 epimer. In fact, Rogers/Molinski only partially assigned the stereochemistry of FS-3 and a C-11 epimer - they did assign C-11, but were not able to assign C-17 in either compound. So it is not true that "C-11 and C-17 have not been assigned," (C-11 has) and it is also not true that Rogers/Molinski fully determined the absolute stereochemistry (they were unable to assign C-17). This whole section should be rewritten to better reflect reality, and underscore that what delineates this manuscript from Molinski's work is determining the stereochemistry at C-17.
Lines 54-56 - The authors cite as Reference 9 - Floren et al and erroneously describe this manuscript as the isolation of 11-epi-Fistularin 2. Typo aside, the Floren paper actually describes the isolation of (+)-1(R),6(S),1'(R),6'(S),11(R),17(S)-fistularin 3, which they also term "(+)-11(R), 17(S)-fistularin-3 or RS-FS3" Given that Floren et al have conclusively determined the C-17 stereochemistry of an "11-epi" compound, this should be made more clear and better attributed in connection with the section above. It would also be better for the authors to use nomenclature which describes the C-11/C-17 stereochemistry more thoroughly when that information is available. The Floren Ref 9 gives much more thoughtful attention to the stereochemical nuances than is presented here. Notably, this second-to-last discussion paragraph of the Floren paper clearly underscores why they themselves are hesitant to call the RS-FS3 compound that they isolated "11-epi-fistularin 3" and so it does them a disservice to contradict this and describe their isolation in this way. This manuscript would be much improved if there was similar attention paid to clearly and accurately stating what is known and unknown about previously-reported isomers of fistularin, as it was more clearly presented in the Floren ref.
Lines 94-96 - Given the above issues, the attempt made here to compare chemical shifts of C-10 and C-17 to Floren's "11-epi-fistularin 3" (again, it is actually "11(R), 17(S)-fistularin 3" - they seemingly did not commit to their compound being the same compound isolated by Molinski) and draw conclusions from them based on their differences is extremely confusing. What is meant by "suggesting different configurations?" What differs from from what, and in what way? I get the impression that comparisons that the authors are attempting to make here would be much easier to follow if the above issues were clarified.
Other minor issues:
Line 45 - Change "Although the isolation of fistularin-3 has been isolated several times" to "Although fistularin-3 has been isolated several times.
Line 61 - Should change to fistularin-3
Line 452 - should this be "electronic CD" instead of "electric CD"?
Author Response
Dear reviewer,
We sincerely thank you for your valuable comments and points. We best revised our manuscript based on your comments and colored the parts in red.
Lines 46-49 - The statements here are self-contradictory. Regarding fistularin-3, is first stated that "its absolute stereochemistry has not been determined because the absolute configurations of the secondary carbinol groups C-11 and C-17 have not been defined," and in the next sentence it stated that Rogers/Molinski determined the absolute stereochemistry of the compound and its C-11 epimer. In fact, Rogers/Molinski only partially assigned the stereochemistry of FS-3 and a C-11 epimer - they did assign C-11, but were not able to assign C-17 in either compound. So it is not true that "C-11 and C-17 have not been assigned," (C-11 has) and it is also not true that Rogers/Molinski fully determined the absolute stereochemistry (they were unable to assign C-17). This whole section should be rewritten to better reflect reality, and underscore that what delineates this manuscript from Molinski's work is determining the stereochemistry at C-17.
→ We rewrote the part as you commented in the revised version. (lines 45-62 on page 2/17)
Lines 54-56 - The authors cite as Reference 9 - Floren et al and erroneously describe this manuscript as the isolation of 11-epi-Fistularin 2. Typo aside, the Floren paper actually describes the isolation of (+)-1(R),6(S),1'(R),6'(S),11(R),17(S)-fistularin 3, which they also term "(+)-11(R), 17(S)-fistularin-3 or RS-FS3" Given that Floren et al have conclusively determined the C-17 stereochemistry of an "11-epi" compound, this should be made more clear and better attributed in connection with the section above. It would also be better for the authors to use nomenclature which describes the C-11/C-17 stereochemistry more thoroughly when that information is available. The Floren Ref 9 gives much more thoughtful attention to the stereochemical nuances than is presented here. Notably, this second-to-last discussion paragraph of the Floren paper clearly underscores why they themselves are hesitant to call the RS-FS3 compound that they isolated "11-epi-fistularin 3" and so it does them a disservice to contradict this and describe their isolation in this way. This manuscript would be much improved if there was similar attention paid to clearly and accurately stating what is known and unknown about previously-reported isomers of fistularin, as it was more clearly presented in the Floren ref.
→ We tried to describe our new stereoisomer by comparison with the known compounds. This explanation was added in the revised version. (lines 45-62 on page 2/17 and lines 445-448 on page 15/17)
Lines 94-96 - Given the above issues, the attempt made here to compare chemical shifts of C-10 and C-17 to Floren's "11-epi-fistularin 3" (again, it is actually "11(R), 17(S)-fistularin 3" - they seemingly did not commit to their compound being the same compound isolated by Molinski) and draw conclusions from them based on their differences is extremely confusing. What is meant by "suggesting different configurations?" What differs from from what, and in what way? I get the impression that comparisons that the authors are attempting to make here would be much easier to follow if the above issues were clarified.
→ We revised the chemical shifts for our FS-3. After analyzing the data again, we found our mistakes for chemical shifts in Table 1. The difference in chemical shifts was slightly small, but could give some meaning. We commented this point in the revised version. (lines 96-106 on page 3/17)
Line 45 - Change "Although the isolation of fistularin-3 has been isolated several times" to "Although fistularin-3 has been isolated several times.
→ Revised
Line 61 - Should change to fistularin-3.
→ done
Line 452 - should this be "electronic CD" instead of "electric CD"?
→ done
Reviewer 3 Report
The work entitled "Stereochemical determination of fistularindds isolated from the marine sponge E. acervus and their regulatory effect on intestinal inflammation" shows the potential therapeutic use of this marine sponge and its metabolites as.
The results are interesting but there are some points that should be clarified.
Why are these three concentrations of the compounds chosen to perform the tests? The IC50 was not determined in each cell line.
Why is it always co-cultivated Caco-2 and THP-1?
The authors should explain why to induce inflammation they apply to cells LPS and + IFN y.
The authors indicate that each compound was added into the apical compartment, and LPS + IFN, was added to the basolateral compartment of the Caco-2 / THP-1 co-culture system. This essay should be explained.
Author Response
Dear reviewer,
We thank you for providing the helpful and insightful comments to improve our manuscript. Our point-by-point response to your comments is provided below. The modified parts are marked in Red. Also, we notice that the manuscript has been carefully edited for spelling, grammar, and readability using Editage’s language editing service.
The work entitled "Stereochemical determination of fistularindds isolated from the marine sponge E. acervus and their regulatory effect on intestinal inflammation" shows the potential therapeutic use of this marine sponge and its metabolites as.the results are interesting but there are some points that should be clarified.
- Why are these three concentrations of the compounds chosen to perform the tests? The IC50 was not determined in each cell line.
→ In cytotoxicity test using CCK-8 assay, compounds 1~6 showed mild cytotoxicity over 20 uM (5 uM for compound 2). Based on this result, 20 uM, the highest concentration without cytotoxicity was chosen, and serially diluted concentrations 10 and 5 uM were set. Also, the exact IC50 value cannot be calculated from the present activity data obtained from three points of concentration. As described in Methods and Materials, the amount of compounds obtained from E. acervus (4.5 kg) is 1.3~4.4 mg (except for compound 3) which is not enough to conduct all assays with five points of concentration.
- Why is it always co-cultivated Caco-2 and THP-1?
→ Our research team has been conducting national R&D projects for about 4 years. The goal of this project is to discover marine sponges and organisms with potent efficacy to treat inflammatory bowel diseases. In accordance with the experimental method in research proposal, in vitro co-culture system of Caco-2 and THP-1 cells has been applied. Accordingly, this paper and three papers previously published by our team in which the same in vitro model was used, are the results of continuous research of the project. Our team is not trying to develop a new in vitro model that resembles intestine with inflammation using different types of epithelial cells and immune cells.
- The authors should explain why to induce inflammation they apply to cells LPS and + IFN r.
→ Interferron-gamma and LPS are generally used stimuli to activate macrophages. Muller et al., (Frontiers in Immunology, 2016) reported that the treatment of IFN-r was shown to synergize with TLR agonist, LPS for induction of macrophage to produce NO and pro-inflammatory cytokines (TNF-a, IL-12). In our co-culture system, the concentration of IFN-r and LPS was optimized to activate THP-1 macrophages based on the production of NO and cytokines, and the expression of inflammatory mediators. The application of LPS+ IFN-r was additionally described in the text.
- The authors indicate that each compound was added into the apical compartment, and LPS + IFN, was added to the basolateral compartment of the Caco-2 / THP-1 co-culture system. This essay should be explained.
→ LPS (10 µg/mL) and IFNγ (10 ng/mL) were added to the apical compartment, and compounds were added into the basolateral compartment. The reasons of the treatment was additionally described in section 3.8.
Round 2
Reviewer 3 Report
The manuscript has been improved and the authors have satisfactorily answered the questions posed. Therefore, I believe that the work could be accepted in the present form.